# Proposal of MUAC as a fast tool to monitor pregnancy nutritional status: results from a cohort study in Brazil

Maria J Miele,[1] Renato T Souza,[1] IMP Calderon,[2] Francisco Feitosa,[3] Debora F Leite,[4] Edilberto Rocha Filho,[4] Janete Vettorazzi,[5] Jussara Mayrink,[1] Karayna G Fernandes,[1] Matias C Vieira ![ORCID],[6] Rodolfo C Pacagnella,[1] Jose Guilherme Cecatti ![ORCID],[1] Preterm SAMBA study group

¹Obstetrics and Gynecology, State University of Campinas, Campinas, Brazil
²Department of Gynecology and Obstetrics, UNESP Campus de Botucatu, Botucatu, Brazil
³Obstetric Department of MEAC, Federal University of Ceara, Fortaleza, Brazil
⁴Obstetrics and Gynecology, Federal University of Pernambuco, Recife, Brazil
⁵Obstetrics and Gynecology, Federal University of Rio Grande do Sul, Porto Alegre, Brazil
⁶Division of Women's Health, King's College London School of Life Course Sciences, London, UK

**Correspondence to**
Professor Jose Guilherme Cecatti; cecatti@unicamp.br

## ABSTRACT

**Objective** In Brazil, although the assessment of maternal nutritional status is recommended using body mass index (BMI), this is only possible in settings adequately prepared. Midupper arm circumference (MUAC) is another biological variable identified as a tool for rapid assessment of nutritional status that is correlated with BMI. Therefore, we aim to surrogate BMI by MUAC cut-offs for rapid screening of maternal nutritional status starting at midpregnancy.

**Design** Analysis of the multicentre cohort study entitled 'Preterm SAMBA' using an approach of validation of diagnostic test.

**Setting** Outpatient prenatal care clinics from five tertiary maternity hospitals from three different Brazilian regions.

**Participants** 1165 pregnant women attending prenatal care services from 2015 to 2018 and with diverse ethnic characteristics who were enrolled at midpregnancy and followed in three visits at different gestational weeks.

**Primary and secondary outcome measures** Sensitivity, specificity, positive and negative predictive values, likelihood ratio and accuracy of MUAC being used instead of BMI for the assessment of nutritional status of women during pregnancy.

**Results** We found a strong correlation between MUAC and BMI, in the three set points analysed (r=0.872, 0.870 and 0.831, respectively). Based on BMI categories of nutritional status, we estimated the best MUAC cut-off points, finding measures according to each category: underweight <25.75 cm (19–39 weeks); overweight 28.11–30.15 cm (19–21 weeks), 28.71–30.60 cm (27–29 weeks) and 29.46–30.25 cm (37–39 weeks); and obese >30.15 cm (19–21 weeks), >30.60 cm (27–29 weeks) and >30.25 cm (37–39 weeks) per gestational week. Therefore, we defined as adequate between 25.75–28.10 cm (19–21 weeks), 25.75–28.70 cm (27–29 weeks) and 25.75–29.45 cm (37–39 weeks) of MUAC.

**Conclusion** We conclude that MUAC can be useful as a surrogate for BMI as a faster screening of nutritional status in pregnant women.

## Strengths and limitations of this study

► This study was a multicentre cohort performed in different regions of the country and exploring changes in maternal body composition.
► The assessment was standardised carried out by maternity health teams and included in the system in real time.
► Although the sample size is robust, the original study was not specifically designed for this assessment, being an analysis of secondary objectives.

obesity can be extended to maternal underweight and malnutrition, with short-term and long-term consequences.[2] The Institute of Medicine (IOM) developed a guideline for American pregnant women to monitor maternal weight gain based on the WHO body mass index (BMI) classification.[3] Public health system in Brazil is monitoring the pregnancy nutritional status by the weight gain, when information on prepregnancy weight is available, or by the BMI charts of Atalah according to gestational week.[4] However, it is necessary to measure maternal weight and height early in pregnancy. In the absence of this information, self-reported weight should be used.[5] In low-income or middle-income countries, it is common for women to seek antenatal care very late. It is also not uncommon for women to skip antenatal care altogether, resulting in missing information on weight before or during pregnancy. As a result, monitoring of nutritional status by BMI becomes less effective.[6–8] Moreover, there is a tendency to underestimate weight and overestimate height when measures are dependent on maternal recall. Anthropometric measurements are frequently not recorded in the prenatal booklet, compromising the accuracy of nutritional status assessment.[9–11]

## INTRODUCTION

Antenatal care represents a window of opportunity to promote healthy life habits, provide individualised care and prevent adverse health outcomes.[1] Worldwide, the concern about

Midupper arm circumference (MUAC) has been recognised as a fast tool adopted to monitor nutritional status, and it is strongly correlated with BMI.[12] It has already been implemented for adult care in under-resourced settings, and for monitoring maternal undernourishment and fetal growth.[13–15] Thus, MUAC allows the assessment of protein intake and storage, related to severe undernutrition.[16] Despite the global obesity pandemic coexisting with undernutrition, there is a maternal and fetal health risk of developing non-communicable diseases such as hypertension, diabetes and delivery of preterm small-for-gestational age neonates, compromising the health of the future offspring.[17 18] Poverty and malnutrition are strongly associated with low birth weight and stunted growth in the first 1000 days of life.[18–21] Brazil has a big variation on demographic density, with higher densities in the South/Southeast regions and Northeast coastal compared with the Northern and Central-West regions with large empty spaces occupied by the Amazon rainforest and turned to agriculture and livestock.[22 23] In this country with socioeconomic disparities, we have an opportunity to investigate in a representative sample an option to faster screening risk of maternal nutritional status from midpregnancy onwards, allowing risk assessment related to abnormal nutrition, which may affect pregnancy outcomes. Implementing an easy and reproducible tool to monitor nutritional status is a great opportunity for scaling-up risk assessment related to abnormal and suboptimal nutritional conditions, monitor pregnancy nutritional interventions and facilitate the provision of a more equitable antenatal care.

## MATERIAL AND METHODS
### Study design
This study corresponds to a secondary objective of the multicentre cohort study entitled 'Preterm SAMBA—Preterm Screening and Metabolomics in Brazil and Auckland', including 1165 Brazilian nulliparous women with singletons, and no history of previous severe clinical condition. Detailed information on research methods and procedures of this study has already been fully explored and published elsewhere.[24 25] Data were collected between 2015 and 2018 at five Brazilian hospitals located in three different geographical regions that have distinctive physical and ethnic characteristics (Maternity Hospital of the University of Campinas (CAISM) and Maternity Hospital from Botucatu Medical School in the Southeast; Maternity of the Hospital of Clinics, Federal University of Rio Grande do Sul in the South; and Clinics Hospital, Federal University of Pernambuco and Maternity School Assis Chateaubriand of the Federal University of Ceará in the Northeast). Pregnant women receiving routine prenatal care who met the inclusion criteria were included in the study between 19 and 21 weeks of gestation for the first study interview, in which data collection and anthropometric measurements were recorded (figure 1).

### Study procedures
The coordinating centre for this study was the CAISM, where the first research meeting took place with health professionals from each centre. These professionals were trained by a nutritionist to take anthropometric measurements in pregnant women. Anthropometric measurements were obtained at three set points: 19–21 weeks, 27–29 weeks and 37–39 weeks. We excluded from the analysis data from women with stillbirth, preterm birth and from those considered as loss of follow-up. All centres inserted maternal anthropometry data into the database of the study (MedSciNet AB, Sweden). Height (cm) and weight (kg) were measured once in each of the three scheduled study visits by a stadiometer and weighing scale, with the feet close to the heels, buttocks and shoulder blades aligned and the head positioned in the horizontal plane of Frankfurt. From the results of height and weight, an automatic estimate of BMI was generated by software using the formula: $weight/height^2$. The MUAC was measured with a flexible non-extensible tape. The left arm circumference was measured in centimetres at the midpoint between the acromion and shoulder by trained health professionals using a standard scale. All MUAC anthropometric measurements were performed three times and the mean was recorded. Some cases from the early beginning of the study had no MUAC measurements, while the team was not enough trained to measure it. Two researchers checked all MUAC results. When the variation exceeded the average of the values of the three measures, the mean of the two closest recorded measures was used. BMI values were categorised by the Atalah curve,[4] based on the Brazilian Ministry of Health's reference to monitor pregnancy,[26] identify adult maternal nutritional status and develop MUAC cut-off points. According to this criterion, the Atalah's classification for 'Adequacy' was used as lower risk for malnutrition and the other categories were considered more likely to develop malnutrition. The adolescents of this sample were 13% and, considering that there is not a specific tool for assess this group,[27] all women were analysed by the Atalah's charts. This instrument was used to identify nutritional status in pregnant women according to BMI and develop MUAC cut-off points. Sociodemographic characteristics, including maternal age, skin colour and family income were self-reported and included into electronic platform of the study.

### Statistical analysis
To compare BMI, MUAC and maternal weight measurements and investigate whether differences in values exist for the measures, a repeated measures one-way analysis of variance test was used. Statistical significance was determined using the Bonferroni's multiple comparisons test, with alpha=0.05. Comparisons were made between three set points (at different gestational ages). For categorical variables, $\chi^2$ tests were performed to analyse differences between BMI classifications at each set point. Each row was analysed individually, without assuming a consistent

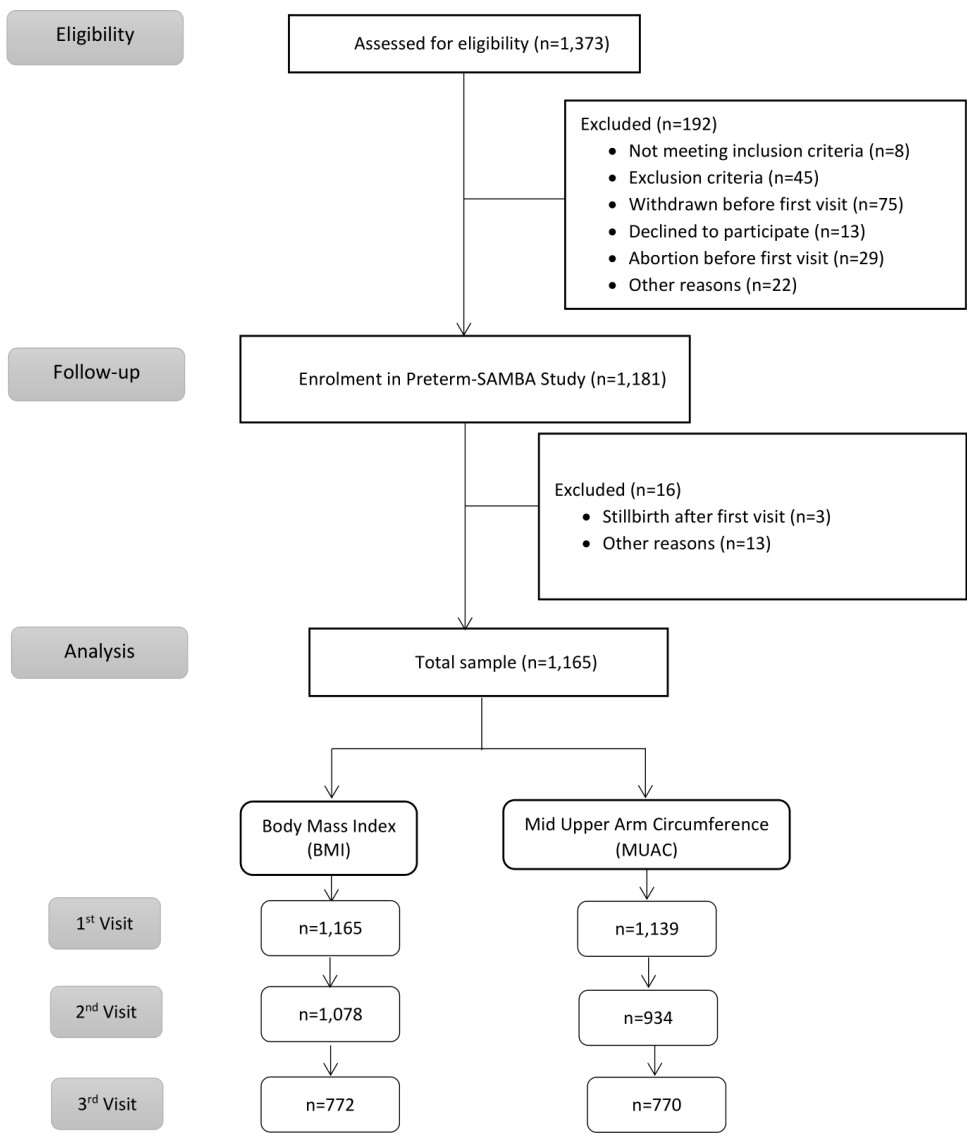

**Figure 1** Flowchart of women participating in the study. Preterm SAMBA, Preterm Screening and Metabolomics in Brazil and Auckland.

SD. Then, a post hoc test was run to assess the differences between all values and find the result of each comparison, using Dunn's multiple comparisons test. Histograms of maternal MUAC and BMI were constructed to determine the distribution of each measurement for each set point separately and for all datasets combined. The normal QQ plot normality and log normality tests were applied using the Shapiro-Wilk test and the Kolmogorov-Smirnov test to compare normal and lognormal distributions. Spearman's correlation coefficient was applied to estimate the linear relationship between MUAC and BMI. Afterwards, the receiver operating characteristic (ROC) curve was plotted to obtain the performance of MUAC measures to predict maternal BMI for all possible cut-off points. The area under the curve (AUC) was calculated along with its 95% CI. Based on coordinates of the curve, the most accurate MUAC cut-off point was selected for the MUAC variable. For each BMI cut-off, we calculated a 2×2 table showing the cross-tabulation of MUAC measurement and

calculated accuracy using McNemar's test of concordance (continuity corrected).[28] To estimate sensitivity, specificity, negative predictive value (NPV) and positive predictive value (PPV), BMI was calculated for each gestational week considering normal the results from BMI adequacy classification. From this categorisation, MUAC values were selected according to BMI references. The Youden method was calculated (highest sensitivity + specificity − 1) to identify the best cut-off measures based on the largest vertical distance between the ROC curve and the diagonal curve, for each visit and separated for classification. The positive and negative likelihood ratio values was calculated according to Fletcher.[29] For the agreement between BMI and MUAC measurements, weighted Cohen's Kappa coefficient was used. Ordinal responses with disagreements were calculated according to their squared distance from the perfect agreement (squared weights).[30]

This study follows the Declaration of Strengthening Report on Observational Studies in Epidemiology.[31]

## Patient and public involvement

Patients or the public were not involved in the design, or conduct, or reporting or dissemination plans of our research.

## RESULTS

Out of a total of 1165 nulliparous women (figure 1), 461 (39.6%) were from the Southeast, 139 (11.9%) from the South and 565 (48.5%) from the Northeast of Brazil. Maternal age at the first visit corresponded to 291 (25.0%) women younger than 20 years (97 under age 17 years), 796 (68.3%) aged 20–34 years and 78 (6.7%) older than 34 years. From the total sample, 578 (49.6%) women self-declared as having brown skin, 117 (10.0%) as having black skin, 462 (39.7%) as white skin and 7 (0.6%) as yellow and 1 (0.1%) as having another ethnicity. Of this sample, 52 (4.5%) women lived with less than US$250 per month, 253 (21.7%) received from US$251 toUS$500;

380 (32.6%) women had a monthly income ranging from US$501 to US$1000 and 480 had an income over US$1000 (41.2%).

Table 1 shows the distribution of anthropometric measurements, including the BMI of all participating women according to each set point. The mean variation in BMI measurements throughout pregnancy was greater than MUAC, and the opposite was observed in extreme measures (minimum and maximum), with greater variations in MUAC and lower BMI.

Figure 2 shows the graphic correlation between BMI and MUAC according to each pregnancy set points measured at the prenatal visits. BMI categories were represented by four different colours and dot size differentiated maternal weight.

To determine the most accurate MUAC value, different categories of nutritional status were selected by BMI classification obtained at three standard prenatal visits scheduled. Differences were significant (p<0.05) for all categories and for three set points analysed (p<0.0001). The AUC was statistically positive at three set points, and

**Table 1** Distribution of the anthropometric measures for each prenatal visit and differences along the gestation

| Characteristics | 19–21 (weeks) | 27–29 (weeks) | First to second visit P value* | 37–39 (weeks) | Third to first visit P value* |
|---|---|---|---|---|---|
| BMI (kg/m²) | | | | | |
| Mean ± SD | 26.33±5.36 | 27.96±5.36 | <0.0001 | 29.93±5.34 | <0.0001 |
| 25th Percentile | 22.50 | 24.20 | | 26.30 | |
| 50th Percentile | 25.40 | 27.00 | | 29.00 | |
| 75th Percentile | 29.20 | 30.70 | | 32.60 | |
| Total (n) | 1165 | 1078 | | 772 | |
| Categories n (%) | | | | | |
| Underweight | 206 (18) | 136 (13) | <0.0001 | 98 (13) | 0.0002 |
| Adequate | 461 (40) | 363 (34) | | 275 (36) | |
| Overweight | 299 (26) | 392 (36) | | 219 (28) | |
| Obese | 199 (17) | 187 (17) | | 180 (23) | |
| MUAC (cm) | | | | | |
| Mean±SD | 28.71±4.64 | 29.32±4.54 | <0.0001 | 29.66±4.35 | <0.0001 |
| 25th Percentile | 25.20 | 26.00 | | 26.50 | |
| 50th Percentile | 28.00 | 28.90 | | 29.00 | |
| 75th Percentile | 31.50 | 32.00 | | 32.00 | |
| Total (n) | 1139 | 934 | | 770 | |
| Weight (kg) | | | | | |
| Mean±SD | 67.95±14.69 | 72.43±14.64 | <0.0001 | 77.65±14.75 | <0.0001 |
| 25th Percentile | 57.60 | 62.33 | | 67.50 | |
| 50th Percentile | 65.40 | 70.00 | | 75.10 | |
| 75th Percentile | 75.70 | 80.08 | | 85.88 | |
| Total (n) | 1165 | 936 | | 772 | |

*ANOVA test for repeated measures. $\chi^2$ test for BMI categories. From the 3015 MUAC measures, 8 values were considered a deviation for those seen at the second visit, the mean of the two closest measures recorded was used.
ANOVA, analysis of variance; BMI, body mass index; MUAC, midupper arm circumference.

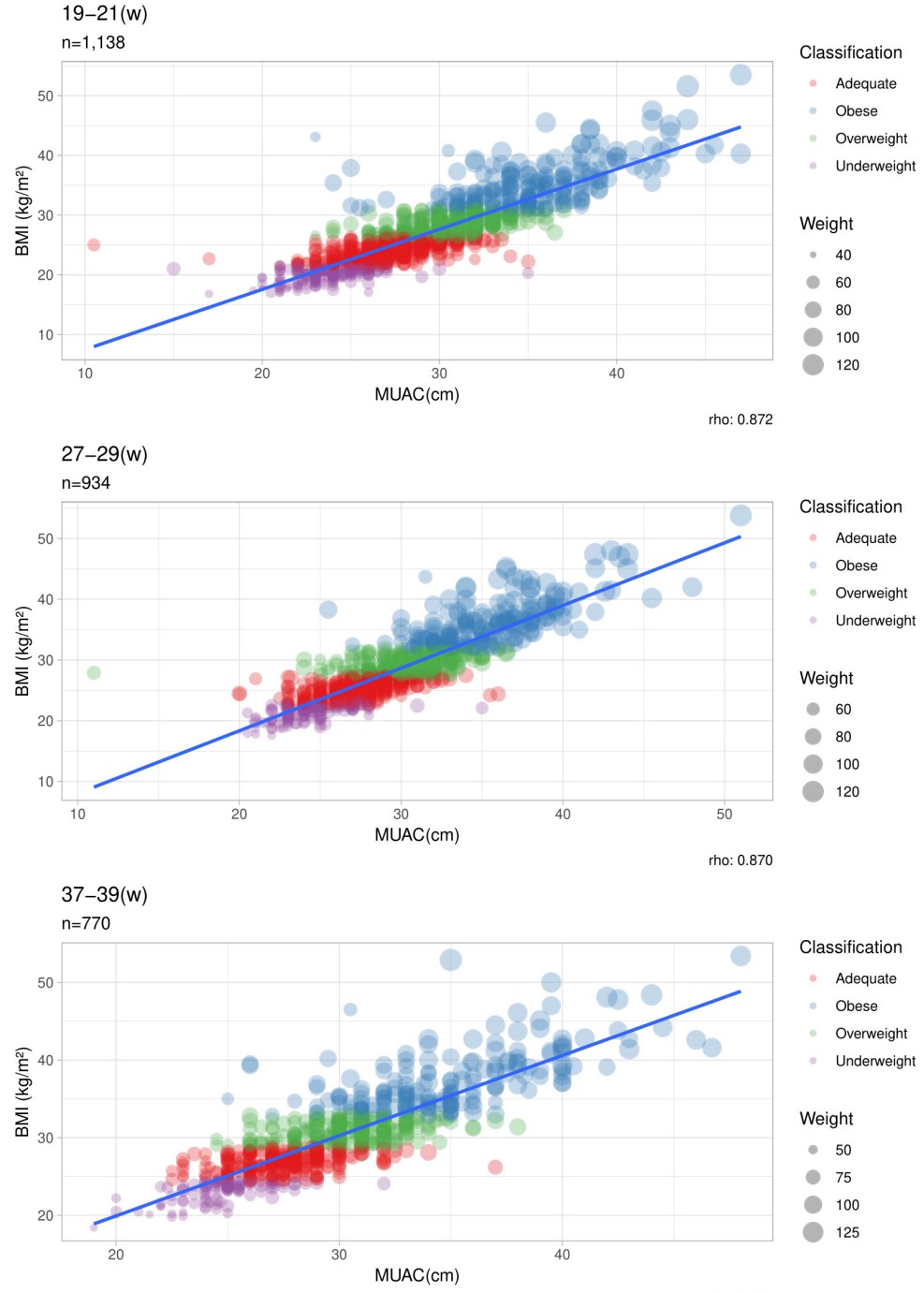

**Figure 2** Correlation between BMI and MUAC for each visit (19–21 weeks, 27–29 weeks and 37–39 weeks). Dot colour indicates the BMI classification. The size of the dots represents maternal weight (kg). Classification: measurements were taken from BMI cut-off values at each week. Rho is the linear correlation coefficient. Statistical Spearman's correlation test. 19–21(weeks): 95% CI=0.8567 to 0.8855, p<0.0001. 27–29 (weeks): 95% CI=0.8532 to 0.8854, p<0.0001. 37–39 (weeks): 95% CI=0.8070 to 0.8522, p<0.0001. BMI, body mass index; MUAC, midupper arm circumference.

robustly significant for screening extreme nutritional status, notably for obese women (figure 3).

The AUC showed that MUAC had a high discrimination capacity for nutritional status obtained by the actual gold standard method (BMI). Table 2 shows MUAC cut-off measures analysed for use as a screening method for maternal nutritional status. At the first prenatal visit, accuracy was lower than at other gestational ages assessed.

It was noteworthy, however, that a substantial difference existed between the predictive values, although the calculus of likelihood (positive and negative) confirmed a slight probability of screening women at risk for malnutrition (obese, underweight or overweight). There was no probability of screening for the lack of risk. The diagnostic test indicated that this measure was a good alternative to the screening test, especially for obese and

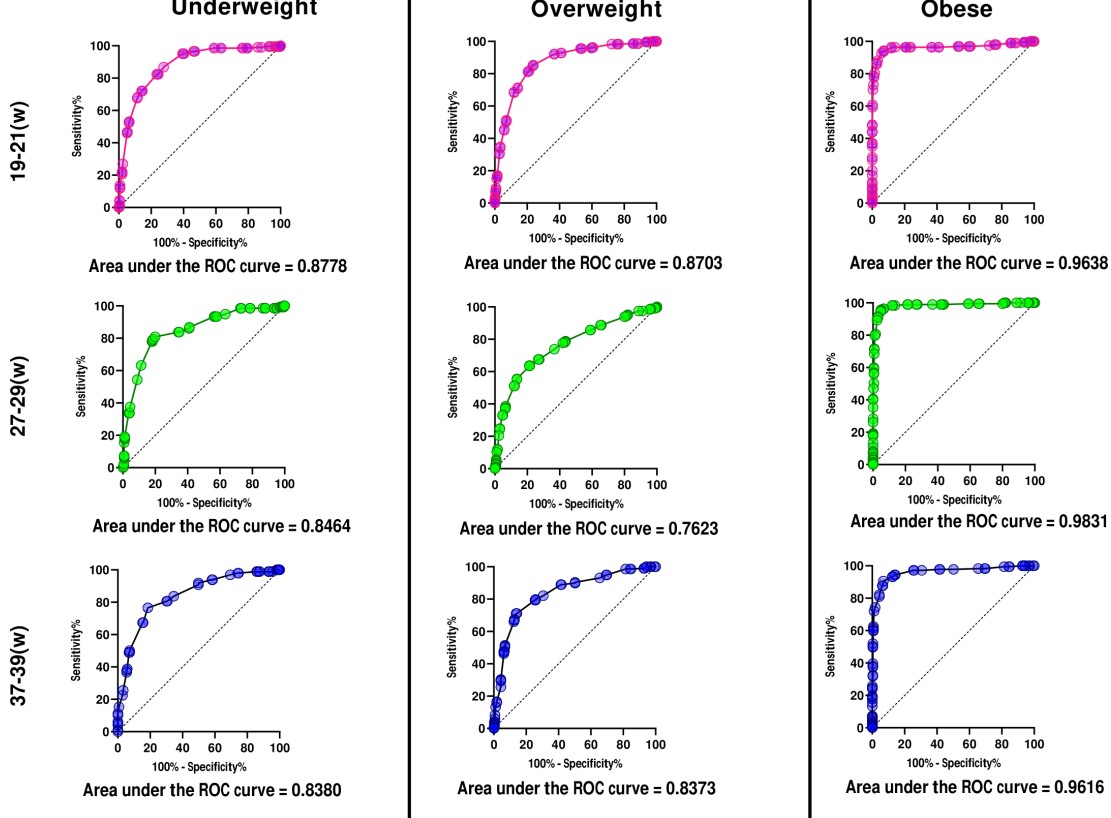

**Figure 3** ROC curves for different MUAC Cut-off values based on BMI categories and the respective area under the curve at three different set points during pregnancy. MUAC, midupper arm circumference; ROC, receiver operating characteristic.

underweight women. Measures between the cut-off for underweight and overweight were identified as within normal, or desirable. Afterwards, an interaction charter between MUAC and BMI classifications was constructed, using selected MUAC measures as a model for gestational age. It is possible to observe differences among the curves, with a better agreement for extremes of classification (obese and underweight). Figure 4 shows changes after the second set point or from the 27th week of gestation onwards, with a better relationship between categories. Comparing the concordance between categories of the two different methods of assessment, a substantial agreement was shown reporting an excellent agreement beyond chance (figure 4). Table 3 presents as a result of our proposal to monitor the maternal nutritional status by cut-offs for the MUAC tape with colours according to the classifications and degrees of risk.

## DISCUSSION
This study investigated the hypothesis that the MUAC cut-off, at different stages of pregnancy, should be a feasible alternative to BMI and is a quicker tool for screening maternal nutritional status in Brazilian pregnant women. We confirmed a strong correlation between BMI and MUAC of Brazilian pregnant women at three set points covering distinct gestational weeks: midpregnancy (19–21 weeks), early third trimester (27–29 weeks) and late pregnancy (37–39 weeks). We achieved our goals of

identifying MUAC cut-off values for different gestational weeks and malnutrition risk levels and present a simple tool as an option to faster screening maternal nutritional status from midpregnancy to 39 weeks.

Our results are similar to findings obtained by studies that have tested this correlation at early and late gestational age, with diverse sociodemographic characteristics.[6 32 33] Most studies have associated MUAC with maternal malnutrition linked to underweight, obesity and low birth weight (LBW). In addition, some authors have assessed MUAC in non-pregnant women and correlated those cut-offs with adult BMI in their own population.[8 34–36]

Effects of inadequate nutritional status during pregnancy increase the risk for low birth weight, preterm birth[37] and small or large for gestational age, among others.[38] Therefore, MUAC has been used as an anthropometric measurement in routine antenatal care to detect malnutrition focused on low maternal weight in middle-income and low-income countries.[39] We identified that a single MUAC measure of <25.75 cm is a predictor of underweight and poor nutritional risk. However, a literature review addressing which anthropometric values indicative of acute malnutrition were associated with adverse birth outcomes, proposed a cut-off point of 23 cm to identify undernourished women. This value was considered to be correlated with lower prepregnancy BMI measures (<18.5 kg/m$^2$) and the occurrence of adverse pregnancy outcomes. Our thresholds are in accordance with the Brazilian recommendation for BMI at

**Table 2** Results of the diagnostic properties of MUAC screening for maternal nutritional status

| Categories | MUAC cut-off (cm) | Sensitivity % (95% CI) | Specificity % (95% CI) | PPV% | NPV% | Likelihood ratio (LR+) | Likelihood ratio (LR−) | ACC% | N |
|---|---|---|---|---|---|---|---|---|---|
| First visit (19–21) | | | | | | | | | 1138 |
| Underweight | <25.75 | 88.06 (82.57 to 92.05) | 65.73 (60.76 to 70.38) | 56.91 | 91.46 | 3.0 | 0.14 | 73.31 | 311 |
| Adequate | ≥25.75; ≤28.10 | – | – | – | – | – | – | – | 281 |
| Overweight | >28.10 | 72.46 (64.10 to 79.55) | 69.58 (63.58 to 75.00) | 55.56 | 82.81 | 2.3 | 0.43 | 70.58 | 180 |
| Obese | >30.15 | 98.92 (95.78 to 99.81) | 60.39 (55.75 to 64.85) | 50.00 | 99.29 | 2.5 | 0 | 71.41 | 366 |
| Second visit (27–29) | | | | | | | | | 1076 |
| Underweight | <25.75 | 82.09 (74.32 to 87.97) | 77.39 (73.08 to 81.20) | 53.14 | 93.26 | 4 | 0.25 | 78.51 | 207 |
| Adequate | ≥25.75; ≤28.70 | – | – | – | – | – | – | – | 356 |
| Overweight | >28.70 | 50.21 (43.69 to 56.73) | 84.52 (80.62 to 87.78) | 64.67 | 75.05 | 2.5 | 0.63 | 72.14 | 184 |
| Obese | >30.60 | 99.44 (96.47 to 99.97) | 70.30 (66.07 to 74.21) | 54.41 | 99.72 | 3.3 | 0 | 77.95 | 329 |
| Third visit (37–39) | | | | | | | | | 770 |
| Underweight | <25.75 | 77.32 (67.48 to 84.95) | 82.26 (77.44 to 86.25) | 57.69 | 92.06 | 3.5 | 0.38 | 81.08 | 130 |
| Adequate | ≥25.75; ≤29.45 | – | – | – | – | – | – | – | 277 |
| Overweight | >29.45 | 42.86 (33.36 to 52.88) | 91.16 (87.16 to 94.03) | 63.38 | 81.71 | 4 | 0.67 | 78.45 | 71 |
| Obese | >30.25 | 94.74 (89.93 to 97.41) | 67.34 (62.45 to 71.88) | 55.48 | 96.75 | 3 | 0.14 | 75.57 | 292 |

Fisher's exact test method which was statistically significant (p<0.05) for all categories at three set points analysed (p<0.0001).
ACC, accuracy; NPV, negative predictive value; PPV, positive predictive value.

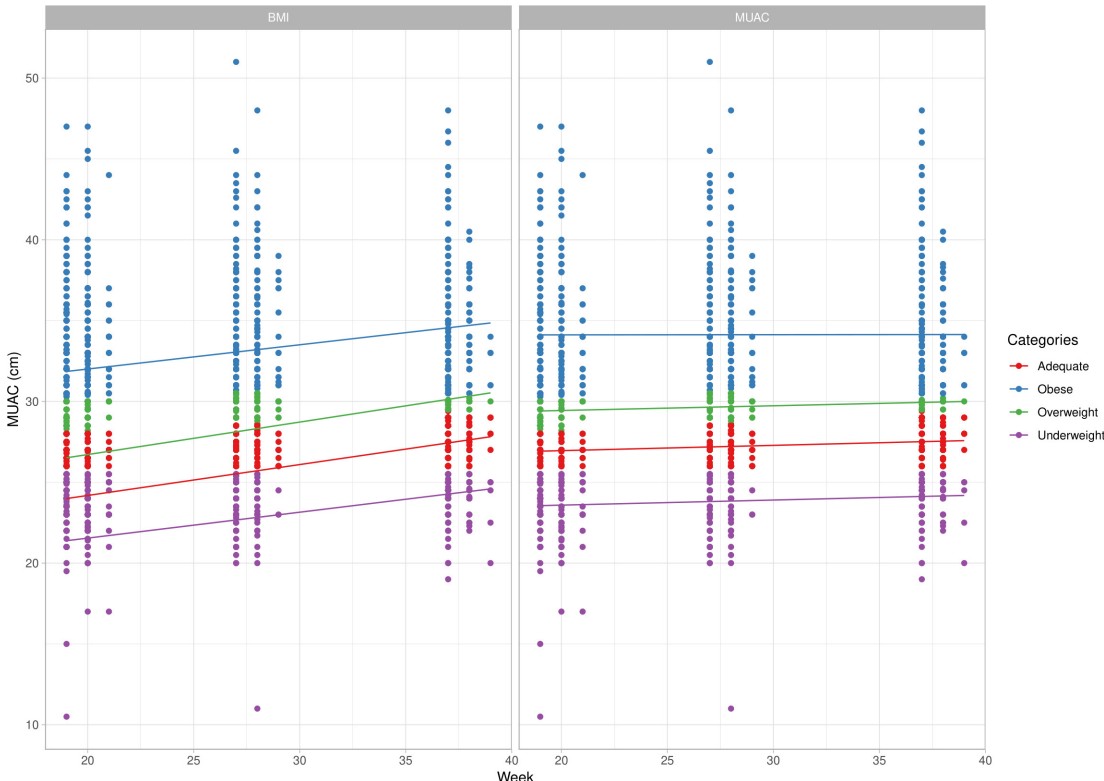

**Figure 4** Agreements between MUAC and BMI, according to MUAC categories by elected cut-off values. In colour, the classification is inside boxes (MUAC and BMI). Lines show the MUAC cut-off for each gestational age. The result of the degree of concordance of Cohen's Kappa according to classification at three set points are illustrated below. Kappa agreement test. 19–21 (weeks): Kappa=0.787, z=24.5, p<0.0001. 27–28 (weeks): Kappa=0.768, z=26.6, p<0.0001. 37–38 (weeks): Kappa=0.765, z=21.5, p<0.0001. BMI, body mass index; MUAC, midupper arm circumference.

midpregnancy. The reference value of the Health System for Brazilian pregnant women at 6 weeks of gestation is a BMI <19.9 kg/m$^2$, close to the IOM recommendations of a BMI of <19.8 kg/m$^2$.[3 40]

Although MUAC is considered a stable measure, with less variation in comparison to BMI,[17] a slight increase on MUAC curves was clearly observed throughout pregnancy. There was a smooth MUAC curve contrasting to the increasing sharp trace of the BMI curve, with a significant agreement among assessments. Our results show not only changes in measures associated with poor nutritional status, but also a mild change in other classifications. A

prior study with women in Argentina tested MUAC cut-off points using the curves generated in their second trimester of gestation to detect LBW. The authors indicated a slight difference in MUAC cut-offs for under-nutrition status according to gestational week, as well as a MUAC of 24.5 cm for pregnant women ≤16 weeks, <25.5 cm at 28 weeks and <26.5 cm at 36 weeks.[6] In India, MUAC was assessed in the first trimester to detect LBW, with an accuracy of 58.7%, as well as PPVs and NPVs of 38.8% and 76.8%, respectively. The authors defined <22.5 cm as the best MUAC cut-off that alerts to an under-weight measure and is a predictor of LBW in the first

**Table 3** Tape colour suggested that screening the probability of risk levels for pregnancy complication based on four MUAC cut-offs (cm) for Brazilian pregnant women at three different gestational periods

| Colour | 19–21 (weeks) | 27–29 (weeks) | 37–39 (weeks) | Interpretation |
|---|---|---|---|---|
| Red | <25.75 cm | | | Underweight |
| Green | 25.75–28.10 | 25.75–28.70 | 25.75–29.45 | Adequate |
| Yellow | 28.11–30.15 | 28.71–30.60 | 29.46–30.25 | Overweight |
| Red | >30.15 | >30.60 | >30.25 | Obesity |

Underweight = risk of poor nutrition, indicate support of diet and nutritional counselling.
Adequate = possibly no risk of poor nutrition.
Overweight = alert to the need for monitoring weight gain, quality of diet and excessive calories.
Obesity = risk for adverse outcome of conditions associated with obesity including high blood pressure, gestational diabetes mellitus and large for gestational age fetus.
MUAC, midupper arm circumference.

trimester.[41] Our AUC values indicate a good diagnostic accuracy of this test between BMI and MUAC. The results showed a high degree of accuracy of MUAC cut-offs satisfactorily beating the current standard of classification for nutritional status (table 2). In terms of correlations between values of MUAC, those results from India are similar to ours, showing a better capacity to find more women who are not at risk, but not missing any women at risk for malnutrition. Nevertheless, in the absence of a 'gold standard' for nutritional risk screening, an African study using a combination of validity tools to diagnose malnutrition found a significant association between MUAC, BMI and the Malnutrition Universal Screening Tool (MUST), searching for poor nutritional status in the adult population (men and women). However, BMI and MUST missed undernutrition in 38% and 43% patients, respectively, whereas MUAC was tracked in 100%. In South Africa, a MUAC value <23 cm was adopted for severely underweight (BMI <16 kg/m$^2$). A value of <24 cm (BMI <18.5 kg/m$^2$) was used to screen for malnutrition or a population at risk for malnutrition.[42]

In our study, a MUAC value of 25.75 was used to identify low maternal weight (19–21 weeks), different from the value proposed in the literature, referring to early gestation or a different population. Although there was a better agreement among values between <22 cm and <23 cm to identify LBW, a MUAC of <23 cm is more highly recommended for the African and Asian contexts, noting that these values were not linked to gestational age.[39] The Khadivzadeh study with healthy women of reproductive age, indicated that a value <24 cm was the best MUAC measure to detect underweight Iranian women.[34] A study in Spain with 1373 male and female subjects of different ages indicated that a MUAC ≤22.5 cm correlates more closely with BMI <18 kg/m$^2$.[35]

The Guidelines for Maternity Care in South Africa monitor maternal nutritional status in antenatal screening using MUAC measures of <23 cm, categorised as underweight. For the group of underweight women, this requires a search for infection and neoplasia. Fetal growth assessment and nutritional support are necessary.[43] A study on the effect of obesity, diet and pre-eclampsia conducted in Ethiopia adopt MUAC classification values <23 cm for underweight, and between 23 cm and <25 cm for adequate. Those authors observed that women with a MUAC <25 cm had healthier habits and ate more vegetables and fruits. Along with the group of MUAC 21–23 cm, those women have a threefold lower chance of developing pre-eclampsia. Women with MUAC between 23 cm and 24.99 cm were considered to have optimal status and <23 cm was categorised as underweight.[17] Similarly, our first cut-off measure that is likely to determine diseases linked to overweight/obesity such as pre-eclampsia, starts at a MUAC of 28.11 cm. It is important to highlight the difficulty in screening for disorders in pregnant women in the first trimester of pregnancy in some countries. The rate of the first antenatal care visit in the first trimester of pregnancy was estimated to be 58·6% worldwide, whereas

it was 48·1% in developing countries and 24·0% in low-income countries.[7]

For the conditions associated with excessive weight (overweight and obesity), South African Guides to maternal care classify a MUAC ≥30 cm as obese. This measurement raises concern for hypertension, pre-eclampsia and gestational diabetes, guiding actions to prevent fetal macrosomia and complications during labour and delivery.[43] In Ethiopia, a MUAC ≥25 cm is categorised as overweight and obese, where authors highlight that MUAC measurements between 28 cm and 39 cm are more capable of predicting complications such as pre-eclampsia.[17] These parameters originated from a previous study in Zimbabwe with a black African population. Those authors found that women with a MUAC between 28 and 29 cm had a fourfold higher risk than those who had a MUAC between 21 and 23 cm.[44]

From a sample of 2000 women, ranging from 15 to 45 years of age in the Islamic Republic of Iran with a BMI >29 kg/m$^2$, it was determined that a MUAC measure >30.5 cm can estimate BMI, detect nutritional disorders and is useful to search for obesity.[34] Previously, a meta-analysis using data from individual participants in seven countries explored the potential to find a significant MUAC cut-off point to identify malnutrition in pregnant women. Mean MUAC varied widely between countries evaluated, as well as according to the weeks in which measurements were taken. In early pregnancy, measurements were between 21.8 cm and 23.0 cm. In the second and third trimester of pregnancy, averages were 25.2 cm and 26.5 cm. In the postpartum period, the average ranged from 27.1 cm to 28.9 cm. The authors concluded that MUAC cut-off values can vary widely internationally and were dependent on the population and time of assessment.[12]

There seems to be a wide difference in MUAC according to the population, age of pregnant women and gestational week for the underweight category (21 cm to 24 cm), and particularly in the search for the overweight/obesity category (29.0 cm to >33 cm). We presumed that nutritional status was adequate at measurements between >24 cm and <29 cm. MUAC cut-off values found in the literature were commonly comparable to prepregnant BMI, with an extremely small difference between those values and our results at midpregnancy. The consequences of extreme anthropometric measurements are linked to adverse outcomes for the mother and child. The socioeconomic, educational and ethnic diversity of the Brazilian population is tremendous, and these characteristics can influence maternal body composition.[45]

MUAC must be specific to different populations. The originality of the study was to investigate a cohort of over a 1000 pregnant women from different regions of Brazil. In the Brazilian pregnant population, there are no studies showing that MUAC is effective at screening risks for extreme cut-offs measures (low weight and overweight/obesity).

We assessed a tool that, to be applied, uses only a tape and can be adopted by obstetricians in the prenatal routines, which is capable of alerting to the risks of inadequate nutrition and their probable consequences linked by the levels of malnutrition, especially needed in settings where anthropometric measures are limited or missing. Once any measure indicating inadequacy is tracked, the need for investigation and eventual support of the diet and nutritional counselling, in addition to monitoring blood pressure and blood glucose, is highlighted.

A possible limitation is that this analysis was conducted as a secondary objective of the original study that had no specific sample size calculation. Therefore, the number of women included could be insufficient for all the associations assessed. All the maternities involved in this study are referral hospitals receiving women from nearby small towns, and therefore, we had a higher rate of loss to follow-up of the prenatal visits with their corresponding measurements. Strengths include prior training of health professionals, real-time data inclusion, use of additional statistical procedures to explore results and mainly, a MUAC cut-off point defined for a Brazilian population of pregnant women (at three regions of the country, with different regional characteristics).

## CONCLUSION

In conclusion, MUAC has the advantages over BMI due to simplicity of application. It does not require the calculation of other measures and is independent of prepregnancy weight recall. In a multiethnic population, our MUAC cut-off measures could be used as a faster screening tool of nutritional status in pregnant women. It may track risk and provide timely intervention to avoid adverse pregnancy outcomes. Still, more research will be welcome in order to validate these cut-offs for pregnancy outcomes.

**Collaborators** The Preterm SAMBA study group: Maria L Costa, Mary A Parpinelli, Rafael B Galvão, José P Guida, Danielly S Santana, Bianca F Nicolosi, Daisy Lucena, Denise F Cordeiro, Elias F Melo Junior, Danilo Anacleto, Lucia Pfitscher, Luiza Brust.

**Contributors** MJM, RTS, JM, RCP, MCV and JGC designed the study. MJM, RTS, JM, IMPC, FF, DFL, ERF, KGF and JV conducted data collection. MJM, JGC and MCV conducted data analysis. All authors had access and participated in the interpretation of results. MJM wrote the first draft of the manuscript, reviewed initially by JGC and then by all authors who read, revised and approved the final version submitted for publication.

**Funding** This study was selected for sponsoring from the research call 'Grand Challenges Brazil: Reducing the burden of preterm birth' number 05/2013 jointly issued by the Brazilian National Research Council (CNPq) (Award 401636/2013-5) and the Bill and Melinda Gates Foundation (grant OPP1107597).

**Disclaimer** The funders played no role whatsoever in the study design, writing of the manuscript nor in the decision to submit the manuscript for publication.

**Competing interests** None declared.

**Patient consent for publication** Not required.

**Ethics approval** The current study was approved by each local Institutional Review Board (IRB) and amended by the Brazilian National Committee for Ethics in Research (CONEP)—Letter of approval 1.048.565 issued on 28 April 2015. The study complies with national and international regulations for human being experiments, including resolution CNS 466/12 of the Brazilian National Heath Council and the 1989 Declaration of Helsinki. All women signed an informed consent form before enrollment.

**Provenance and peer review** Not commissioned; externally peer reviewed.

**Data availability statement** Data are available upon reasonable request.

**ORCID iDs**
Matias C Vieira http://orcid.org/0000-0002-8076-4275
Jose Guilherme Cecatti http://orcid.org/0000-0003-1285-8445

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
