## [Reviewer comments · BMJ Open]

ARTICLE DETAILS

TITLE (PROVISIONAL)	The proposal of MUAC as a fast tool to monitor pregnancy nutritional status: results from a cohort study in Brazil
AUTHORS	Miele, Maria; Souza, Renato; Calderon, IMP; Feitosa, Francisco; Leite, Debora; Rocha Filho, Edilberto; Vettorazzi, Janete; Mayrink, Jussara; Fernandes, Karayna; Vieira, Matias; Pacagnella, Rodolfo; Cecatti, Jose

VERSION 1 – REVIEW

REVIEWER	Bhatia, Vikas All India Institute of Medical Sciences - Bhubaneswar
REVIEW RETURNED	24-Feb-2021

GENERAL COMMENTS	A proposal of a fast tool to monitor pregnancy nutritional status (submitted to BMJ) Comments 1. Abstract: The results section appears cluttered with too many details. Simplified presentation might help in better understanding Use MESH terms for keywords. 2. Introduction: Overall language improvement needed in this section. 3. Methods: The authors need to provide justification why BMI was used as a gold standard for comparison against MUAC. 4. Results: There was a loss of follow up of > 20% during the study. The authors need to comment on it and it can be mentioned as one of the limitations. 5. Discussion: None 6. Tables: Please provide denominator for each table. 7. References: none
--

REVIEWER	Mosquera, Paola
-----------------	-----------------

	Universidade de São Paulo
REVIEW RETURNED	18-Mar-2021

GENERAL COMMENTS	Dear authors, Prior to publication, some points need to be reviewed. The general comments below may guide the authors: Abstract:  - Please, could you better explain the following sentence? “In Brazil, the monitoring is by calculating Body Mass Index (BMI) using curves categorized with nutritional status.” - Please, make it clear what is “w”. Introduction:  - Lines 63 to 66: Please, see if you can be more specific. The IOM recommends optimal weight gain based on pre-pregnancy BMI categories and the Brazilian Ministry of Health adopted these recommendations, in combination with Atalah’s curves (Brazil 2012). Later lines are confused if those details are not clarified. - The information presented from line 84 to 86 (“Brazil has a different demographic density...”) I do not think is necessary. Methods:  - What do CAISM and SD mean? Please, be careful with the acronyms. Is university of Campinas the meaning of CAISM? - In Figure 1: what do you mean by “other reasons”? Why do you have a smaller n for MUAC than for BMI on each visit? -How many times height and weight were measured in each visit? -Kindly, could you rewrite the following sentence for further understanding (line 124)? “When the average variation was considered a deviation from the mean of the remaining values from the same women, the mean of the two closest recorded measures was used.” - Line 126: “BMI values were categorized by...., identify adult maternal nutritional status and develop MUAC cut-off points.” What happened with the teenagers? What was the classification used for the BMI of adolescents? - Could you rewrite the following sentence for further understanding (line 128)? “According to this criterion, the adequacy of Atalah’s classification determined that there was no risk for malnutrition and not the adequacy of other classifications.” Results:  -The following variables: maternal age, self-declared skin color and income were not explained in the methods section. -In the text, it is not clear what figure 2 reveals. Could you please give us a brief explanation? Also related to figure 2: “The Shapiro-Wilk normality was used to test whether data were different from normal distribution ($p < 0.05$). Spearman’s correlation coefficient was applied.” I consider that this information is not appropriate in results section. -The indication “(Figure 2)” is not necessary. It is already clear at the beginning of the paragraph. -The following sentence was previously detailed in the statistical analysis section: “To compare the concordance between categories of two different methods of assessment, Cohen’s Kappa statistic was applied” -Table 1: is 90(%) percentile correct? -Table 2: please, check if the “Wilson-Brown method” was described in the methods section.
--

	-Tables: please, be careful with the acronyms. Discussion:  - Please, could you support the following sentence with bibliographic references? “Malnutrition during pregnancy increases the risk for low birthweight, preterm birth, and small or large for gestational age, among others.” -The following sentence is not clear: “Although MUAC is considered a stable measure, with less variation in comparison to BMI17, it was clearly observed between both curves by the trace throughout pregnancy.” Improve the English. -Regarding “In India, MUAC was assessed..., with an accuracy of 58.7,...” Is it percentage? - I suggest avoiding the following in the discussion section: (table 2). Please, make sure you explain it in the results section. About: “High AUC values, in most results, show a good relationship between BMI and MUAC.” Are you referring to this study? Concerning “In terms of correlations between values of MUAC, those results are similar to ours...” Which ones? Please, review the sentence from line 249 to 253 (page 10). -Limitation of the study: losses to follow-up? -The sentence from line 327 to 332 (page 12) is repeated in the Conclusion section.  -Please, check all the references and make sure the institutions names are consistently cited. (Brasil, BRASIL, Brasil MS?) -Overall: I recommend editing the text to achieve a more appropriate grammar structure and scientific language. Please, review punctuation marks throughout the text.
--	--

VERSION 1 – AUTHOR RESPONSE

Reviewer: 1

Dr. Vikas Bhatia, All India Institute of Medical Sciences - Bhubaneswar

Comments to the Author:

Comments

1. Abstract:

- The results section appears cluttered with too many details. Simplified presentation might help in better understanding

We apologize for the text and have improved a few changes. We hope to be clearer now.

- Use MESH terms for keywords.

Agreed, we have updated the terms.

2. Introduction: Overall language improvement needed in this section.

Ok, the whole text was double checked by a skilled professional.

3. Methods:

- The authors need to provide justification why BMI was used as a gold standard for comparison against MUAC.

In Brazil, the Ministry of Health uses the BMI charts combined with IOM orientation for weight gain. However, the monitoring of the pregnant nutritional status is performed using the Pregnant Woman's booklet which has the Atalah's charts. Following the reviewer suggestion, we agreed with you and have included this information both in the Abstract and in the Introduction sessions.

4. Results:

- There was a loss of follow up of > 20% during the study. The authors need to comment on it and it can be mentioned as one of the limitations.

This was a real cohort where the pregnant women were not necessarily following prenatal care in the same clinic where data was collected. Therefore unfortunately there was a rate of loss to follow up higher than we would like. We acknowledged and included this information as a limitation aspect in the discussion.

5. Discussion: None

6. Tables:

- Please provide denominators for each table.

Thank you for that suggestion. We have inserted the required information in the tables.

7. References: none

Reviewer: 2

Dr. Paola Mosquera, Universidade de São Paulo

Comments to the Author:

Dear authors,

Prior to publication, some points need to be reviewed. The general comments below may guide the authors:

Abstract:

- Please, could you better explain the following sentence? "In Brazil, the monitoring is by calculating Body Mass Index (BMI) using curves categorized with nutritional status."

We are agreed and made the change for "In Brazil, the Pregnant Woman's booklet uses the Body Mass Index (BMI) charts for monitoring the pregnant nutritional status".

- Please, make it clear what is "w".

This was an incorrect abbreviation for "week". We now completed the information and corrected for "week".

Introduction:

- Lines 63 to 66: Please, see if you can be more specific. The IOM recommends optimal weight gain based on pre-pregnancy BMI categories and the Brazilian Ministry of Health adopted these recommendations, in combination with Atalah's curves (Brazil 2012). Later lines are confused if those details are not clarified.

We agreed and changed the sentence accordingly. We thank the reviewer for this suggestion.

- The information presented from line 84 to 86 ("Brazil has a different demographic density...") I do not think is necessary.

We had included this sentence to explain to the international audience that different regions of the country have different socioeconomic and therefore nutritional habits possibly involved in the association we are looking for. This to show that we were also thinking on the representativeness of the Brazilian population.

Methods:

- What do CAISM and SD mean? Please, be careful with the acronyms. Is university of Campinas the meaning of CAISM?

Ok, we had completed the information.

- In Figure 1: what do you mean by "other reasons"?

There were multiple reasons in fact, including some obstetric conditions found during pregnancy, declination to continue, moving to another place, etc. Then we decided to summarize with "other reasons".

Why do you have a smaller n for MUAC than for BMI on each visit?

In the very beginning of this study, before the health team from the prenatal care has been fully trained, they did not take the MUAC measurement. However, we had the weight and height for all the women and then decided to not miss that information for calculating BMI as recommended. This is now written in the study procedures.

-How many times height and weight were measured in each visit?

All the anthropometric measurements were assessed once at three different appointment on the prenatal: 19-21, 27-29 and 37-39 gestation weeks. This information was clearly introduced in the study procedures session of the manuscript.

-Kindly, could you rewrite the following sentence for further understanding (line 124)?

“When the average variation was considered a deviation from the mean of the remaining values from the same women, the mean of the two closest recorded measures was used.”

OK, we rewrote the sentence to make the message understandable.

- Line 126: “BMI values were categorized by...., identify adult maternal nutritional status and develop MUAC cut-off points.” What happened with the teenagers? What was the classification used for the BMI of adolescents?

We had 13% of teenagers in the sample, who were also assessed according to the Atalah’s curves considering there are not otherwise recommendations. We now included this information on method.

- Could you rewrite the following sentence for further understanding (line 128)? “According to this criterion, the adequacy of Atalah’s classification determined that there was no risk for malnutrition and not the adequacy of other classifications.”

We appreciate the suggestion and changed the sentence accordingly.

Results:

-The following variables: maternal age, self-declared skin color and income were not explained in the methods section.

OK, we agree with this and included this information.

-In the text, it is not clear what figure 2 reveals. Could you please give us a brief explanation? Also related to figure 2: “The Shapiro-Wilk normality was used to test whether data were different from normal distribution ($p < 0.05$). Spearman’s correlation coefficient was applied.” I consider that this information is not appropriate in results section.

We totally agree and rewrote this part of the text following the suggestions.

-The indication “(Figure 2)” is not necessary. It is already clear at the beginning of the paragraph.

OK, deleted.

-The following sentence was previously detailed in the statistical analysis section: “To compare the concordance between categories of two different methods of assessment, Cohen’s Kappa statistic was applied”

We appreciate the suggestion and changed the sentence accordingly.

-Table 1: is 90(%) percentile correct?

We agree this information does not make sense there and we deleted it.

-Table 2: please, check if the “Wilson-Brown method” was described in the methods section.

Well noted. We also deleted this information.

-Tables: please, be careful with the acronyms.

We apologize. We are thankful for this observation and have made the necessary corrections.

Discussion:

- Please, could you support the following sentence with bibliographic references?

“Malnutrition during pregnancy increases the risk for low birthweight, preterm birth, and small or large for gestational age, among others.”

In fact, it was really missing. Again, we apologize and included the reference.

-The following sentence is not clear: “Although MUAC is considered a stable measure, with less variation in comparison to BMI17, it was clearly observed between both curves by the trace throughout pregnancy.” Improve the English.

We very much appreciate this suggestion. After the appropriate changes, the sentence became much clearer.

-Regarding “In India, MUAC was assessed...., with an accuracy of 58.7,...” Is it percentage?

Yes, it is a percentage. Now the sign % has been included.

- I suggest avoiding the following in the discussion section: (table 2). Please, make sure you explain it in the results section. About: “High AUC values, in most results, show a good relationship between

BMI and MUAC.” Are you referring to this study? Concerning “In terms of correlations between values of MUAC, those results are similar to ours...” Which ones? Please, review the sentence from line 249 to 253 (page 10).

Your suggestion was plainly accepted and, after clarifying your idea and provide the information that was missing, the meaning was better reported.

-Limitation of the study: losses to follow-up?

We agreed with this and informed as one limitation as well, also following the suggestion of the other reviewer.

-The sentence from line 327 to 332 (page 12) is repeated in the Conclusion section.

Apologies. This was an error. The first paragraph was deleted.

-Please, check all the references and make sure the institutions names are consistently cited. (Brasil, BRASIL, Brasil MS?)

Done.

-Overall: I recommend editing the text to achieve a more appropriate grammar structure and scientific language. Please, review punctuation marks throughout the text.

Thank you for all your kind comments and we had provided the adjustments indicated.

Reviewer: 1

Competing interests of Reviewer: None declared

Reviewer: 2

Competing interests of Reviewer: None declared

VERSION 2 – REVIEW

REVIEWER	Bhatia, Vikas All India Institute of Medical Sciences - Bhubaneswar
REVIEW RETURNED	28-Apr-2021

GENERAL COMMENTS	Thank you for making amendments as per my comments
--

REVIEWER	Mosquera, Paola Universidade de São Paulo
REVIEW RETURNED	07-May-2021

GENERAL COMMENTS	The authors made several improvements in the text. The points I raised have been sufficiently addressed. I consider that the article is now suitable for publication in this Journal.
---